# Sports Organizations and Their Defensive Mediatization Strategies: The Sports Journalist's Perspective

**Neil O'Boyle * and Aaron Gallagher**

School of Communications, Dublin City University, D9 Dublin, Ireland; aaron.gallagher62@mail.dcu.ie
* Correspondence: neil.oboyle@dcu.ie

**Abstract:** This article provides empirical evidence of 'defensive mediatization strategies' in the field of sport. These are strategies used by actors individually and collectively to control and sometimes avoid media publicity—for example, by refusing requests for media interviews, or in the case of an organization, by making media literacy training available to its staff. In this article, we use the concept of defensive mediatization strategies to identify and illuminate some of the challenges facing professional sports journalists in the postbroadcast era. The article draws on findings from an ongoing study of the relationships between professional sports organizations, athletes, and journalists, but reports only on interviews conducted with experienced sports journalists in Ireland and Britain (*n* = 16). Our analysis identifies a number of defensive mediatization strategies used by sports organizations, including increased levels of in-house media, differential treatment of journalists, and an increasingly competitive stance towards journalism generally. We also consider a potentially more pernicious strategy: the hiring of professional sports journalists as internal communications advisers—a switching of role positions that might be termed 'poacher turned gamekeeper'. The article organizes findings according to the three categories of defensive mediatization strategies identified in the extant literature (*persistence*, *shielding*, and *immunization*) and proposes a fourth category, which we label *steering*.

**Keywords:** sports journalists; sports organizations; defensive mediatization; Ireland; Britain

## 1. Introduction

What sports journalism is, how it should be defined, and how this particular area of work within the news industry is being shaped by emerging technologies are matters of ongoing debate, discussion, and investigation within the sports studies field. If digital networked technologies are widening and altering the nature of participation in sport (by fans, athletes, etc.), they are also transforming how sports journalism is produced, circulated, and consumed. For example, Frandsen et al.'s (2022) notion of 'participatory liveness' points to the centrality of social media in the shaping and coverage of media events, including sporting ones. To some extent, social media enable users to bypass and circumvent traditional journalism and, therefore, might be viewed as competitors to it. For example, Hutchins and Mikosza's (2010) study of the 2008 Beijing Olympics describes how traditional broadcasting strategies at this event collided with the networking capacities of Web 2.0 and 'social software' (blogs, video-sharing sites, and social networking sites), resulting in different and sometimes much more critical assessments of the unfolding event entering the public domain (p. 284). However, Nölleke et al. (2017) argue that suggestions that social media are *competitors* to traditional journalism are too simplistic and that the relationship between sports journalism and social media is largely complementary. For example, they find that the social media accounts of athletes often act as news sources for sports journalists, enabling them to gather 'inside information'.

In recent years, a number of sports scholars have turned to the concept of 'mediatization' as a theoretical entry point for analyzing the direct, indirect, and structural

effects of media. Mediatization describes increasing relations of interdependence across social domains that depend, in large part, on media-related processes. 'Through these relations, the role of 'media' in the social construction of reality becomes not just partial, or even pervasive, but 'deep': that is, crucial to the elements and processes out of which the social world and its everyday reality is formed and sustained' (Couldry and Hepp 2017, p. 62). Mediatization is consequential for all actors, from individuals to largescale organizations, and its effects are sometimes ambivalent or double-edged. For example, Frandsen and Landgrebe's (2022) study of the introduction of the Video Assistant Referee (VAR) to the Danish Superliga draws on the concept of mediatization to reflect on how football 'is becoming ever more closely tied to the technological and institutional logics of media' (p. 816). These authors argue that the introduction of VAR instigated complex processes of change that not only decreased football's institutional autonomy but also created inequalities between larger and smaller leagues within Europe's football structure. On a much smaller scale, Birkner and Nölleke (2016) investigated how athletes perceive media influence and logic and how such perceptions shape their media-related behavior, finding that engagement with media can bring considerable financial reward but also interference in one's private life. Studies have also examined social relations and communicative behavior *around* sports. For example, Skey et al.'s (2018) 'bottom-up' study of the mediatization of football demonstrates how digital networked communications technologies have significantly transformed the ways in which sport is accessed, enjoyed, and participated in—for example, through the use of streaming services, social media, and online gaming. These and other studies make clear that mediatization is an uneven, nonlinear, and multidimensional process (Birkner and Nölleke 2016), and that it does not occur in a uniform way across all social domains and cultural contexts (Frandsen and Landgrebe 2022; Ličen et al. 2022).

To date, studies of the mediatization of sport, including those already mentioned, have documented the transformative effects of this process and the strategies used by actors individually and collectively to adapt to media attention and gain public visibility. For the most part, they describe accommodations *toward* media and generally paint a picture of decreasing institutional autonomy. As Frandsen and Landgrebe (2022, p. 812) put it, 'All processes of mediatization involve negotiations of values, roles, and practices among agents in the field in question; and ultimately, all imply a decrease both in individual and in institutional autonomy' (Frandsen and Landgrebe 2022, p. 812). However, Nölleke et al. (2021) argue that studies of mediatization sometimes overlook or disregard the ways actors *avoid* and *control* media publicity and thereby 'protect established structures and practices against media demands' (2021, p. 738)—what they term 'defensive mediatization strategies'. This concept does not imply wholesale resistance to, or a refusal to engage with, media. Rather, it highlights that mediatization is never a simple one-way process of accommodation and typically entails a mix of offensive and defensive strategies.

Nölleke, Scheu, and Birkner suggest that defensive mediatization strategies are evident in all domains of society. Indeed, one can argue that they are also evident at the level of nation states—for example, when oppressive regimes impose bans on journalists, and disrupt or even 'cut off' internet services (see Lengel and Newsom 2014)—arguably the most extreme defensive mediatization strategy possible. Based on a secondary analysis of data from previous research projects, Nölleke, Scheu, and Birkner describe three categories of defensive mediatization strategies (*persistence*, *shielding*, and *immunization*), which are mutually reinforcing and operate at the levels of individual actors, organizations, and a social system's routines and norms. *Persistence*—or more precisely, persistence in 'pre-mediatized behavior' (p. 746)—describes attempts by actors to push back against the demands of media and persist with established structures and practices. For example, an organization might decide not to invest in public relations even though its competitors are. *Shielding*, as the name suggests, involves active attempts to block media. For example, an organization might simply refuse requests for media interviews. Finally, *immunization* describes efforts to develop capacities in 'handling' the media. For example, an organization

might make media literacy training available to its staff members. Again, it is important to stress that individuals and organizations are never *wholly* defensive in their responses to mediatization: 'In practice, most social actors probably see both benefits and risks to media publicity and public attention and will therefore implement a mixture of offensive and defensive mediatization strategies' (ibid., p. 740).

Like Nölleke, Scheu, and Birkner, our approach to mediatization is *institutionalist*, which is to say that it approaches society as an interinstitutional system and analyzes how media penetrate and shape but do not *colonize* other institutions. As Hjarvard (2014, p. 202) argues, mediatization is a *reciprocal* process and 'concerns the co-development and reciprocal change of institutional characteristics of both media and other domains'. Put differently, this approach acknowledges that institutional structures are both stable and dynamic and that institutions have their own unique 'logics' (rules and resources). Accordingly, an institutionalist approach to mediatization 'allows for an understanding of how the logics of the media intersect with the logics of other institutional domains' (ibid., p. 203). This theoretical orientation is useful because it enables us to consider how actors *proactively shape* mediatization processes, whilst also acknowledging that different actors can have different understandings of how media publicity works (or fails to work).

While defensive mediatization is a relatively new theoretical construct, previous research has identified strategies and actions by sports organizations that align with this perspective and tell us much about the challenges contemporary sports journalists face. For example, Borges' (2019) study of the emergence of club-owned media in the context of European soccer finds that this development coincided with a tightening of press access to players and coaches. Similarly, Suggs' (2016) research on American intercollegiate athletics finds that access to athletes has become more restrictive for journalists and that some have been sanctioned for what sports organizations have perceived as inaccurate or unflattering reporting. His research also finds that some athletic programs and associations have imposed limits on real-time blogging and social media posting on the grounds that such work 'infringes on broadcast rights' (p. 263). Research has also found that many sports organizations are increasing their communications budgets, hiring more media personnel, and channeling increasing funds into maintaining websites, producing social media content, and other media-related activities (Grimmer and Kian 2013; Frandsen 2016; Borges 2019; Mirer 2019). For example, Frandsen (2016) finds that Danish national sports federations now spend almost three quarters of their communication budgets on maintaining websites and social media. Hutchins et al. (2019, p. 981) similarly report that 'a growing number of clubs, leagues and associations are also partnering with video streaming technology providers'. For example, Major League Soccer has established a partnership with Apple TV, while the Premier League has partnered with Amazon Prime. Of particular concern for journalists—as we examine further below—is the significant rise in 'in-house' or 'team' media within sports organizations, a development that, on one hand, bypasses independent media and, on the other, blurs the boundaries between journalism and public relations (English 2022; McEnnis 2021). As McEnnis (2021, p. 10) suggests, team media are ultimately concerned with 'brand image rather than public service'. This can also be seen in the often dull and 'safe' interviews given by athletes and officials, who are in some cases obligated to follow corporate briefs or use specific lines (Sherwood and Nicholson 2017). Studies have also found that public relations practitioners sometimes even attempt to impose restrictions on what *parts* of an interview are published (Grimmer and Kian 2013). Such developments are clear exercises in agenda setting, although this is hardly new. For example, in 2000, Fortunato concluded that 'much of the creation of NBA related mass media content is directed by the NBA *itself*' (p. 481, our emphasis). In this article, we attempt to identify and illuminate some of the challenges facing professional sports journalists in the postbroadcast age while also responding to Nölleke, Scheu, and Birkner's call for empirical studies of defensive mediatization strategies. Such strategies are used by sports organizations to exert more control over media narratives, but also have the effect of creating a less hospitable environment for sports journalists and can ultimately

hinder their ability to perform their role as the Fourth Estate. Evidence suggests that even new videoconferencing technologies, such as Zoom, are being used to limit and control access, with some journalists expressing concerns that the platform is being used even when in-person interviews are possible (Gentile et al. 2022). In what follows, we present findings from an ongoing study of the relationships between professional sports organizations, athletes, and journalists. The larger study includes interviews with all three parties and a documentary analysis; however, our analysis here derives solely from interviews with journalists working in Ireland and Britain. In the conclusion, we reflect on the limitations of such an approach.

## 2. Materials and Methods

Like Frandsen and Landgrebe (2022, p. 818), our primary research interest is in the reflections of 'core actors'—in this case, sports journalists—with a view to providing 'a more nuanced understanding of mediatization' (Nölleke et al. 2021, p. 739). The next phase of the larger study, of which the findings presented here form part, will include interviews with US-based sports journalists; however, our analysis in this article is limited to data garnered from interviews with journalists working in Ireland and Britain ($n = 16$). These proximate states share many characteristics as journalistic cultures and media systems (e.g., a strong public service broadcasting ethos, high levels of professionalization, considerable levels of newspaper circulation, etc.). British media also command a substantial share of the Irish media market; though it is worth noting that Irish media content—especially sports content—is also popular with British audiences (see Dwyer 2020).

The study reported here uses key informant sampling (Oliver 2021)—a form of sampling in which participants are selected based on their specialized knowledge or role. Given his background in sports journalism, the second author drew up a list of potential interviewees, which was reviewed by the first author. The list included only those individuals with five or more years' work experience. We were also careful to include female sports journalists, given the underrepresentation of females in the sports field generally—i.e., as journalists, pundits, managers, coaches, and policy decision-makers (Liston and O'Connor 2020). On this point, it is worth adding that even the sports-related output of public service broadcasters such as the BBC often reinforces this male-centeredness (see Ramon and Rojas-Torrijos 2022). Despite our efforts, only two female sports journalists agreed to be interviewed.

Interviews were carried out between November 2022 and March 2023. Fourteen informants were male, two were female, and all gave their informed consent. Depending on the availability and work schedules of informants, interviews were conducted via Zoom ($n = 14$) or face-to-face ($n = 2$). Interviews followed a semi-structured, topic-orientated format and averaged forty-five minutes in duration. The second author transcribed the interviews and checked them for accuracy, and both authors independently and manually coded the transcripts. Coding of the raw data was partly deductive and guided by Nölleke, Scheu, and Birkner's theorization of defensive mediatization. Like these authors, we paid special attention to interview data that referred 'to the objective of protecting against or avoiding media attention' (Nölleke et al. 2021, p. 744). Coding was also partly inductive—i.e., open and generative of theory. We used reflexive thematic analysis (RTA) to code, categorize, and thematize findings, following Braun and Clarke's (2006, 2019) six-step framework (familiarization with data, generation of initial codes, search for themes, review of themes, defining and naming themes, and production of the report). Along with independent coding, we discussed and compared our respective interpretations and reviewed codes and themes on an ongoing basis—as per Clarke and Braun's instruction that the six steps are recursive and not intended to be completed in a linear fashion. The core task of our thematic analysis was to *reframe* and *reinterpret* informant responses in terms of defensive mediatization strategies (see Kiger and Varpio 2020, p. 3). In doing so, we constructed a number of thematic maps, which enabled us to connect elements in the data and also locate strategies in terms of institutional levels—i.e., micro, meso, and macro.

## 3. Results

In interviews, informants were asked to reflect on sports journalism as a practice and profession in the context of wider changes affecting the news media, to comment on the profession's evolving jurisdiction—or more precisely, on 'jurisdictional disputes' (Covaleski et al. 2003)—and to describe their interactions and relationships with the sports organizations they report on. Our analysis here focuses primarily on the last of these themes, though clearly all are interconnected, and is guided by the concept of defensive mediatization strategies.

Nölleke, Scheu, and Birkner present their findings under three headings—microlevel strategies, mesolevel strategies, and macrolevel strategies—and we follow suit here. Figure 1 shows examples of each category of defensive mediatization strategy at each institutional level. The use of circles (concentric and Venn) is deliberate and indicates that strategies and institutional levels overlap and intermingle. As explained below, our analysis also suggests an additional category, which we label *steering*.

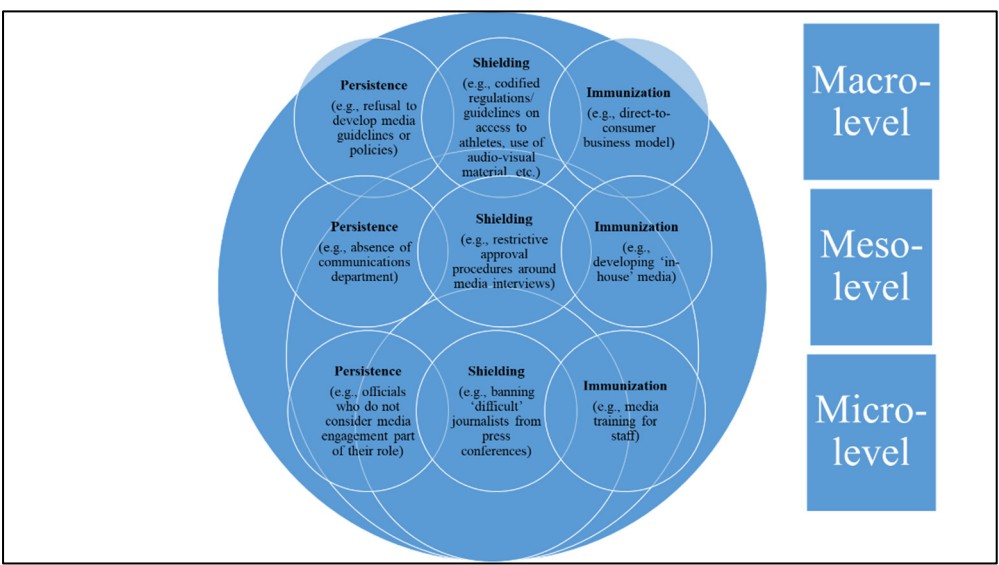

**Figure 1.** Defensive mediatization strategies at each institutional level.

### 3.1. Microlevel Defensive Mediatization Strategies

Interviews with journalists pointed to a number of defensive mediatization strategies operating at the microlevel, though few offered clear examples of *persistence*. Some suggested that while most sports officials are willing to engage with media *on some level*, a small number simply do not view it as part of their remit, which obviously makes the job of journalists more difficult, given their reliance on such gatekeepers. More commonly, informants described efforts by sports organizations to actively block journalists at the microlevel—i.e., *shielding*. Indeed, a journalist at SportsJOE fittingly commented, "Some media managers are what we like to call *blockers* where they want to control absolutely everything." A sports journalist at *The Daily Telegraph* similarly claimed that restrictions around access are often less to do with athletes and coaches and more to do with media managers who "overestimate the importance of their role" or simply wish "to be *seen* to be in charge". Informants also complained that some press officers/media managers ignore their requests for interviews, refuse to respond to their emails, and sometimes offer few if any explanations for why interview requests were rejected. For example, a sports journalist at *The Daily Telegraph* remarked, "I would just like a bit more transparency from sports organizations in terms of us putting in interview requests and being granted access. It can be frustrating when you request an interview and you don't hear back from them, or they don't give you a reason for not granting access." Some also suggested that a journalist who might be perceived as 'difficult' might be banned from a press conference, briefing, or the

mixed zone—an example of what might be termed *disciplining*. For example, a broadcaster at RTÉ Sport (Ireland's public service broadcaster) suggested that the well-known Irish journalist and sports pundit Eamon Dunphy was a "thorn in the side of Jack Charlton" (the Republic of Ireland's football manager between 1986 and 1996)—or a 'persona non grata', to use Suggs' (2016, p. 263) term. The informant claimed that Dunphy "was removed from the press pack", but insisted that "less has changed in that sense than people like to think".

Responses by informants also revealed evidence of *immunization* at the microlevel. *Immunization* describes efforts by individuals and organizations to develop capacities in handling the media, and thereby (it is hoped) avoid or lessen 'dysfunctional consequences' (Nölleke et al. 2021, p. 745). Almost all our informants mentioned media training, especially on the part of athletes. For example, a sports broadcaster at Virgin Media commented, "if a sports organization is reasonably sophisticated in how it handles its media side of things, it will have trained its athletes and players and staff to say as little as possible." Likewise, a sports journalist at *The Irish Times* commented, "Athletes are absolutely much more media trained nowadays. Answers are more bland, they are more careful in what they say . . . The majority of athletes, because of the media training they receive from a very young age, tend to be more corporate in the way they speak".

### 3.2. Mesolevel Defensive Mediatization Strategies

Interviews with journalists pointed to a number of defensive mediatization strategies operating at the mesolevel of departments and resources; however, none of our informants provided clear examples of *persistence* at this level. A possible reason for this is that our research focused on experienced sports journalists, who, in most cases, are accustomed to working with relatively large sports organizations with well-developed media functions.

Informants described various efforts by sports organizations to block journalists at the mesolevel—i.e., *shielding*. For example, a number spoke about increasingly restrictive procedures around the conduct of media interviews and the stifling effect this had on interactions between journalists and athletes. For example, a sports journalist at The Athletic UK claimed that more often than not, there is "a comms officer breathing down your neck" when conducting interviews. He added that journalists are unlikely "to get great quotes in that environment, because everybody is on edge". Likewise, a football columnist at *The Times UK* remarked, "It can be difficult when a media manager sits in on an interview. It happens a lot when you sit down to do a one-on-one feature interview with Premier League footballers, and it doesn't make for the best environment for an interview, in my experience. They do want to control things a lot".

In respect of *immunization*, almost all our informants commented on the rise of 'in-house' or 'team' media. While in one sense, in-house media offer a 'shield' against journalists, arguably their primary function is to bypass external media where possible and develop internal media capabilities. For example, a sports broadcaster at Virgin Media commented, "Sports organizations producing their own content is visibly a part of the landscape now . . . And I think that is a factor in why these sports organizations are being a little bit more uncooperative with us in the traditional print and broadcast media. They realize that they can create their own content, so why would they bother cooperating with us." Similarly, a sports journalist at The Athletic UK remarked, "as journalists, we have now come to expect that when someone at a club wants to say something, they will use the club's in-house media channels to say it". He added that his main concern around this development is that it "impacts how often we can sit down with high profile players and managers, because clubs can do a lot of that content themselves".

### 3.3. Macrolevel Defensive Mediatization Strategies

Our interviews with journalists were of limited use when it came to identifying defensive mediatization strategies operating at the macrolevel of sports organizations. This is perhaps unsurprising. As Nölleke et al. (2021, p. 749) point out, 'self-reports seem rather problematic when it comes to investigating the macro-level of mediatization'. Once again,

our research found little evidence of *persistence* at the macrolevel, presumably because most if not all the sports organizations described by informants had already undergone significant levels of restructuring to accommodate media. Nevertheless, while they did not go so far as to suggest increasing levels of *hostility* towards journalists, a number described what could be interpreted as increasing levels of *shielding*—by large sports organizations in particular. For example, informants described increasingly restrictive rules around schedules, press conferences, the use of audio-visual material, and access to athletes. Some claimed, possibly due to the emergence of in-house media (see above), that some sports organizations are becoming more defensive towards journalists, not because they perceive them as a sort of necessary nuisance but because they view them as *competition*. For example, an online sports journalist at RTÉ Sport commented, "Some sports organizations do view the media as competition for content and stories and clicks and views. Sports teams realize they can produce content themselves and they don't need us as much as before." Likewise, a journalist at SportsJOE commented, "Things have changed in that way where the sports teams that we report on are almost in competition with us for the audience and the views and clicks".

Linked to this, but which is perhaps better considered *immunization*—i.e., building institutional capacity—is a perceived shift towards direct-to-consumer business models within larger sports organizations. As the name suggests, this strategic reorientation involves efforts to develop *direct* relationships with audiences and fans rather than relying on intermediaries. The motivation here is primarily about commercialization, monetization, and data collection, and involves a shift in strategic thinking towards intellectual property and media assets. However, an associated and logical consequence of such an orientation is an increase in the development of in-house content. For example, the cycling correspondent of *The Irish Times* commented, "cyclists have sponsors and sponsors want publicity, but journalists aren't as needed to tell cyclist's stories, because teams can tell stories themselves, directly to fans. So that has cut down on the access we get." Similarly, a sports journalist at *The Times UK* commented:

> "Athletes and sports stars are often treated like assets that are to be protected by the sports organizations and teams they are employed by. A lot of sports teams now see themselves as competitors to media outlets. Clubs have their own media teams and do their own interviews, and as a result of that they are less inclined to give interviews or give us information. The media's role as conduit between clubs and fans is much less than it was before. There used to be a time when the only way a club could connect with its fans was via a newspaper, but that's not the case now."

### 3.4. Steering

Our analysis in the preceding sections offers some empirical support for the defensive mediatization strategies identified by Nölleke et al. (2021). As already established, these authors describe three categories of defensive mediatization strategies (*persistence*, *shielding*, and *immunization*), which are mutually reinforcing and operate across the micro-, meso-, and macrolevels of an organization. However, our analysis suggests another potential category that is related to (and nestled between) *shielding* and *immunization* but is subtly different to both. We have labelled this category *steering*. *Steering* differs somewhat from *shielding* and *immunization* in the sense that it is neither a strategy of merely 'blocking' nor 'developing capacity' but more accurately about 'guiding' media towards desirable ends. Clearly, the development of in-house or 'team' media (see above) is partly motivated by the same aim, but it is also motivated—perhaps more so—by the aims of maximizing revenue, monetizing audiences, and generating data. *Steering*, we suggest, is more narrowly focused on controlling media narratives and is, therefore, a more proactive strategy than *shielding*. Our interviews with journalists suggested a number of examples of *steering* at each of the three institutional levels, as illustrated by Figure 2.

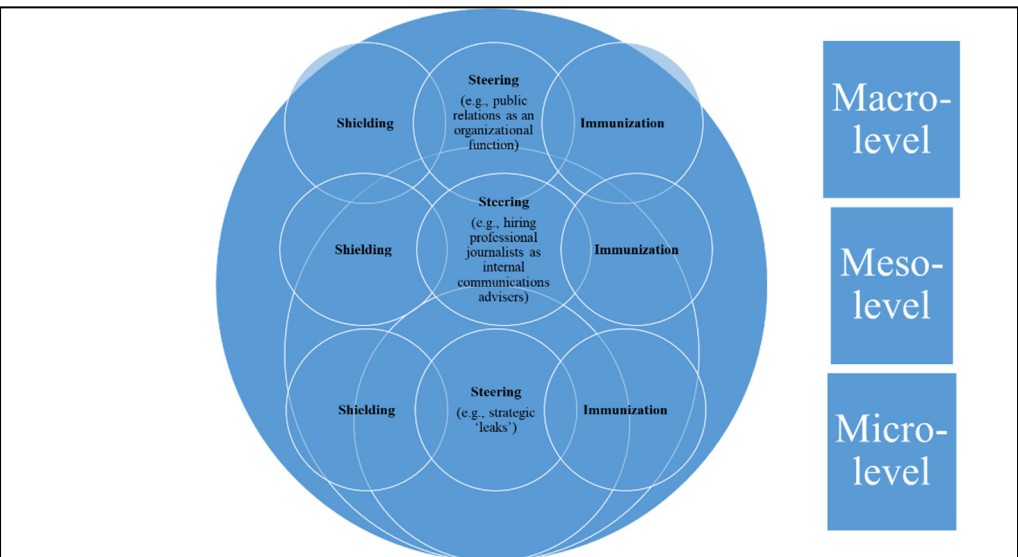

**Figure 2.** *Steering* at each institutional level.

A microlevel example of *steering* is when athletes follow a corporate brief in interviews or are given specific lines to use. For example, a sports broadcaster at Virgin Media described such interviews as little more than "media trained answers". He suggested that they are not only of limited journalistic value but can actually be counterproductive for athletes, who can "come across as quite robotic and lacking personality". The cycling correspondent at *The Irish Times* similarly suggested that such interviews are often highly "sanitized" and result in "boring, bland quotes". Other examples of *steering* at the microlevel include issuing media packs, media releases, and other prepackaged, 'ready to use' information subsidies. On this point, a broadcaster at RTÉ Sport commented, "Sports organizations have become such experts in their communications. The IRFU (Irish Rugby Federation) will craft and perfect a press release to such a high standard nowadays that media publications can almost publish them straight away without the need to edit them or change anything. The easier you make a press release to use, the more likely a journalist will use it, and the more chances it will be used in the way you sent it out from your organization."

Such attempts at agenda setting do not end with the careful crafting of press releases but extend to strategic considerations of *who* should receive them. For example, an online journalist at RTÉ Sport remarked, "Some sports organizations will send information to every journalist, but others are a bit more *selective* over who they send information to. Different sports organizations have different approaches over who gets certain stories". Relatedly—and providing another example of *steering* at the microlevel—is the strategic 'leaking' of information. On this, an online journalist at RTÉ Sport remarked, "It is tricky when you are leaked information, because you know the sports team that has leaked you the information has a certain agenda." A sports writer at *The Irish Times* elaborated on the difficulties such leaks present for journalists:

> "Leaks happen all the time in sports journalism. But there are some journalists who are leaked stories by teams, and they are essentially doing the team's bidding for them, being used by the team. Journalists will get *exclusives* that make them look like fantastic reporters. But, when you look at it closely, the journalist is basically working for the sports team, doing their bidding, working in tandem with them. As a journalist, you get leaked information all the time, but it's what you decide to do with that information that really matters. You have to be aware of the fact that as a journalist, you are being used by the leaker. You have to question their motivation. They want run to run a story that meets their needs and objectives, and you have to be aware of that."

A mesolevel example of *steering*, which to some extent also fulfills the overlapping functions of *shielding* and *immunization* (as indicated in Figure 2), is what informants described as the deliberate hiring of professional journalists as internal communications advisers. This switching of role positions, which we coded 'poacher turned gamekeeper', can be interpreted as an attempt (by the sports organization) to not only harness journalistic expertise and develop internal capacity but also neutralize and *redirect* it—or "control the story", as a broadcaster at RTÉ Sport put it. In a similar vein, a sports journalist at *The Irish Independent* commented, "Who better to manage your media than somebody who works, or worked, in the media themselves? They know how to manage the media and what to prevent. Poacher turned gamekeeper, it's the perfect foil." Some informants expressed a degree of indignation towards such individuals, implying that compromised ethics had pushed them beyond the 'boundary' of professional journalism (see Mirer 2019). For example, a sports columnist at *The Irish Examiner* commented, "Sometimes it does piss people off, because you get the impression that some of them have forgotten their roots as journalists". However, most acknowledged that a journalist's decision to 'switch sides' is often motivated by the simple desire for a more "stable career path", as a sports broadcaster at Virgin Media put it. Nevertheless, the reasons for this switching of role position are potentially manifold—as expressed by a sports journalist at The Athletic UK:

> "I think it's always happened in media. Most club's comms teams are full of former journalists. Many Premier League clubs have former journalists working for them who worked for their club's local papers, or national newspapers too. It happens for a number of reasons. One reason is that they are better paid working for Premier League clubs than local media organizations. Maybe they fancy a new challenge, especially with access diminishing. Maybe a journalist wants to be closer to the actual action".

Another example of *steering* at the mesolevel is the *continued* use of Zoom for press conferences, even though pandemic-related health risks have diminished considerably. Our informants interpreted this as another means of maintaining control and suggested that sports organizations often limit invitations to hand-picked journalists, are selective when answering questions, and sometimes even 'mute' journalists to avoid follow-up questions. For example, a UK-based freelance journalist commented:

> "[Zoom] really interrupts the flow of the press conference and prevents good information and answers being brought forward. It stifles it, and maybe that's the motive and the intention [of the sports organization]. The more interruptions, the more they can control the environment, limit the number of questions, not allow follow-up questions, it all means they can control the message better, from the sports organization's point of view, to the detriment of us as journalists and our audiences and readers too."

Similarly, a multimedia journalist for RTÉ Sport commented:

> "Sports organizations can control things a little bit more, because so many press conferences and interviews are now being held over Zoom, since the Pandemic. It's a lot more convenient to host them that way, but it means that they can choose who attends, who gets to ask questions, and the opportunity to ask follow-up questions isn't there, because on Zoom they can just mute you. If it's an in-person press conference, it's a bit easier to ask a follow-up question and hold people accountable, in that way."

Finally, a macrolevel example of *steering* at a sports organization is the establishment of public relations as an organizational function alongside other functions such as planning, staffing, and so on. Our informants suggested that PR plays an increasingly important role across all sports organizations and that those working in a PR capacity are paid to control the message, to promote positive messages, and to avoid "content that will make their employers *look bad*"—as an online sports journalist at RTÉ Sport put it. However,

the increasing role of public relations and related developments, such as the emergence of in-house media, do not of course *guarantee* that *steering* will be successful. For example, a multimedia sports journalist at RTÉ Sport claimed that audiences and fans are now much savvier about the content they consume:

> "I will watch an interview done with an in-house sports media team, and then see the same player interviewed by an independent media publication or journalist, and I know which one I'm going to prefer. You can tell which interview is the fluff piece and which one is more likely to have a bit of a story behind it . . . I think the public and the audience are still very self aware about what content is authentic and which one is biased, and bland and PR."

## 4. Discussion

Our aims in this article were twofold: first, to develop and further theoretical discussion of defensive mediatization strategies and provide some empirical grounding for future research; and second, to use this theoretical construct as a lens for reflecting on the challenges facing professional sports journalists in the postbroadcast age. The institutional approach followed here posits that as media are integrated into other institutions, *both* sets of logics are transformed. As Hjarvard (2014, p. 219) puts it, 'the particular outcomes of these reciprocal accommodations should be examined empirically, and the logics of the media are certainly not always the most influential'. In the same vein, the concept of defensive mediatization strategies highlights that actors individually and collectively 'not only take measures to attract media attention but also employ strategies to protect against it' (Nölleke et al. 2021, p. 744). In other words, mediatization is not a simple, one-way, or uniform process but more accurately entails offensive *and* defensive strategies. However, Nölleke, Scheu, and Birkner argue that while the offensive and defensive goals of actors often overlap, it is important that we at least try to distinguish them, as this will help us to explain how mediatization processes are *proactively shaped*. We share their view that journalism studies can benefit from 'an extended concept of mediatization that incorporates defensive adaptations' (ibid., p. 753).

Our findings point to a number of challenges facing sports journalists in the postbroadcast era, many of which have already been identified in the extant literature. Some of these challenges relate to the changing nature of work in an increasingly hybrid media environment. For example, all our informants suggested that social media have transformed their work practices and day-to-day routines, if not necessarily their core values. They suggested that social media offer networking and storytelling opportunities but also come with risks, including hostile feedback and misinformation/disinformation, and that the line between content produced by professional journalists and that produced by individuals who may be masquerading as journalists has blurred. This last point has a direct bearing on journalistic claims to professional authority and raises questions about the value and distinctiveness of sports journalism. Indeed, McEnnis (2021, p. 2) argues that sports journalists are experiencing 'fundamental, existential concerns' and that their professional base is now threatened 'by new actors who have adopted its norms, practices, codes, routines and values' (p. 2). More broadly, it suggests that the profession's future sustainability depends not only on continued demand, but also on the audience's ability to differentiate "PR and journalism"—as the online sports editor at RTÉ Sport put it. Informants also highlighted difficulties around maintaining journalistic accountability across an array of work practices (filing stories, live tweeting, posting to Instagram, podcasting, etc.) and new pressures brought on by social media metrics.

In addition to these environmental challenges, our informants spoke at length about the defensive practices of sports organizations, which we analyzed using Nölleke, Scheu, and Birkner's framework. Based on an extensive secondary analysis of previous data, Nölleke, Scheu, and Birkner identify and describe three categories of defensive mediatization strategies. *Persistence* refers to efforts by actors to 'persist in or strengthen already-established structures, regulations, routines, etc. even if they individually or collectively

perceive that the media and the public would prefer if they changed' (p. 744). *Shielding* involves efforts to curtail or avoid media attention. 'To shield against media demands means that actors consciously implement structures in order to minimize public attention' (ibid., p. 744). Finally, *immunization* describes efforts to develop capacities in handling the media and thereby avoid or lessen 'dysfunctional consequences' (ibid., p. 745).

At the microlevel of daily interaction, our informants reported experiences of being blocked, ignored, and in some cases denied access by media managers. Naturally, issues around access were particularly concerning for them, given their reliance on sources to produce work; however, it also raises more fundamental questions about their ability to produce quality, unbiased work. For example, McEnnis (2021, p. 3) observes that the fear of having 'access revoked' puts pressure on journalists and can sometimes lead them to produce more 'unquestioning stories'. Issues of access have also arisen in relation to videoconferencing technologies such as Zoom—a finding that is also reported by Gentile et al. (2022). At the mesolevel—the level of organizational departments and resource allocation—our informants commented on several developments, most notably the rise of in-house or team media at sports organizations. Many suggested that the ability to create their own content and use their own channels and platforms has made sports organizations somewhat less cooperative with independent media—a finding that has been reported in other studies. As McEnnis (2021, p. 3) puts it, 'What used to be a rather balanced relationship, in which clubs and organizations relied on sports journalists for the oxygen of publicity, has given way to a lop-sided power dynamic whereby these gatekeepers now have their own digital and social media channels and are therefore less reliant on independent media'. Our analysis also found evidence of what we call *steering*, which is related to, but different from both *shielding* and *immunization* insofar as it is more narrowly focused on controlling media narratives. Our informants offered a number of examples this, including strategic leaks, information subsidies, hiring professional journalists as internal communications advisers, and increasing levels of selectivity when working with journalists. Again, similar findings have been reported in other studies. For example, Sherwood and Nicholson (2017) find that many of those working in media relations and communications roles within Australian sports organizations hail from journalistic backgrounds. Likewise, they find that when organizing media conferences, some Australian sports organizations are highly strategic and selective when it comes to inviting journalists (see also Suggs 2016). Such occasions are, therefore, not simply about disseminating information but are increasingly 'viewed by professional sport organizations as an opportunity to set the media agenda' (p. 147).

As noted above, our interviews proved of limited use when it came to the macrolevel, though several of our informants remarked that sports organizations were transforming 'from facilitators for media to *competitors* and publishers with a dominant market advantage' (English 2022, p. 856, our emphasis). The absence of commentary on macrolevel strategies is an interesting finding in itself and suggests that in focusing on busy day-to-day tasks and interactions, these Irish and British-based journalists may be less cognizant of macrolevel decisions that have a direct bearing on their work. For example, in the US context, research by Fortunato (2000) on the National Basketball Association (NBA) has shown that rules and policies around access to athletes, media relations training, and other media-related activities are generally part of macrolevel strategies. For instance, he notes that rules on access to players and coaches 'are provided as a league-wide standard timing format' (pp. 487–88). He also notes that media relations training is a mandatory requirement for all players entering the NBA and forms a core component of the Association's Rookie Transition Program.

The research presented here has a number of limitations, foremost of which is that our analysis tells only one side of the story as it were. In other words, as important and illuminating as they are, the views of sports journalists can (and likely will) differ in some respects from those working on behalf of sports organizations. As Johnson (2002, p. 105) observes, interviews often highlight that individuals or groups involved in the same line

of activity can 'have complicated, multiple perspectives' on the same phenomena. It is important to add that interviewing as a research method suffers from a number of limitations and is, therefore, often triangulated with other sources of data. Moreover, the journalist–organization relationship examined here clearly does not exist in a vacuum, and crisscrossing and extending beyond this relationship are the complex, converging practices of multiple agents across a multiplicity of media forms (see Frandsen et al. 2022). Such agents—including fans, citizens, and interest groups—are increasingly willing and able to direct negative feedback or 'flak' (Herman and Chomsky 1988) to all sorts of organizations, including sporting and news-making ones. Finally, it important to note that the study described here focuses on 'sports journalists' but takes little account of individual differences of gender, race, ethnicity, and other categories of social identity—nor does it give adequate consideration to potential differences regarding national contexts, institutional cultures, or wider media systems. We suggest that these sources of potential variation—and how they might affect an individual journalist's 'possibilities to act' (Hjarvard 2014, p. 208)—are given greater attention in future studies of defensive mediatization strategies.

Despite these limitations, our analysis helps to illuminate some of the challenges facing sports journalists in the current conjuncture. Efforts by sports organizations to evade public attention, to avoid or steer media coverage, and to protect autonomous decision-making can make rigorous and quality reporting more challenging and can ultimately make it more difficult for journalists to hold such organizations to account. It is worth adding that 'traditional' journalistic jobs are in increasingly short supply relative to those in the professional sports environment and that many media organizations are under considerable financial pressure. Indeed, given such fraught circumstances, a broadcaster at RTÉ Sport claimed that he was entirely unsurprised that many journalists are leaving the industry to work in sports organizations:

> "I'm not one bit surprised. You get better paid, you have less hassle, and you can control the story without having to go and *look* for a story. And there's a multitude of factors that have caused this trend: the lack of access to players, the drop in salary, the job losses in our sector. Those of us who are left working in sports journalism are basically *survivors* at the moment. The slow death of newspapers and the re-emergence of digital subscriptions ... it's a battle out there for media outlets to stay alive. I'm not surprised that so many people have left journalism."

A larger question that arises from this study is how sports journalists can best navigate these myriad challenges and secure their profession going forward. For McEnnis (2021, pp. 3–4), it is essential that they commit to 'serious journalism', which requires (a) building institutional autonomy from sources; (b) providing depth and rigor in reporting; and (c) ensuring their work is socially responsible and reflective of the interests, lives, and values of diverse communities. McEnnis' insistence on holding power to account and maintaining professional standards is undoubtedly correct; however, as we have seen, defensive mediatization strategies make these goals significantly harder to achieve and ultimately impede, undermine, and obstruct the work of journalists. Tension—and perhaps a degree of mutual suspicion—has always been a vital and necessary element in the relationships between journalists and those they report on. However, our findings suggest that the increasingly competitive and, at times, hostile relationship between journalists and sports organizations is making investigative work much more difficult. As a sports journalist at *The Irish Times* commented:

> "It's an adversarial relationship. Sports teams want media when they can use us for what they need ... When they can use us, the media, then it's a positive relationship for them, but they hate the media actually doing deep, investigative reporting on how they operate. The media are a nuisance to them. It's purely adversarial. They don't want us around if we're going to ask difficult questions. I think sports organizations now relish the fact that they can now control the

message a lot more. They can do a lot more without the media than they ever could before. That, to them, must be heavenly."

**Author Contributions:** Conceptualization, N.O. and A.G.; methodology, N.O. and A.G.; validation, N.O. and A.G.; formal analysis, N.O. and A.G.; investigation, A.G.; resources, N.O. and A.G.; data curation, A.G.; writing—original draft preparation, N.O. and A.G.; writing—review and editing, N.O. and A.G.; visualization, N.O. and A.G.; supervision, N.O. and A.G.; project administration, N.O. and A.G. All authors have read and agreed to the published version of the manuscript.

**Funding:** This research was supported by DCU's Faculty of Humanities and Social Sciences' Staff Journal Publication Scheme 2022–2023.

**Institutional Review Board Statement:** The study was conducted in accordance with the Declaration of Helsinki, and approved by Ethics Committee of Dublin City University (DCU-FHSS-2022-029, 25 May 2022).

**Informed Consent Statement:** Informed consent was obtained from all subjects involved in the study.

**Data Availability Statement:** The interview data presented in this study are not publicly available due to privacy restrictions.

**Conflicts of Interest:** The authors declare no conflict of interest.

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
