# Peer review of "Sports Organizations and Their Defensive Mediatization Strategies: The Sports Journalist’s Perspective"

_journalmedia, doi:10.3390/journalmedia4020042_

Round 1

Reviewer 1 Report

Thank you for the opportunity to review your manuscript, “Sports organizations and their defensive mediatization strategies. The sports journalist’s perspective” for Journalism and Media.

The questions here are obviously massively important for understanding the shifts in the sports-media production complex. The experience of doing sports journalism in this more hybrid media environment has changed the professional experience of the independent journalist as well as the claims he or she makes to professional authority. Understanding that side of the equation is an important contribution to literature in this area. The concept of defensive mediatization is an interesting way to address this question and I think it is a useful way to think about these processes.

I guess my main question about this manuscript is the multi-level framework that the researchers use to classify these strategies. In terms of Section 3.3, it is interesting to me how much the subjects actually cannot see past their daily interactions to see how they are actively shaped by policies set for them. But in this context, does it not seem that everything is an outgrowth of macro-level decisions? Hiring choices, the decisions to launch in-house media, decisions about access policies, and courting of the press all would be part of an overarching strategy. I guess I wonder if the use of mediatization as a framework for thinking about it reinforces the distinctions between sports and media organizations and at the same time as it is asserting that sports organizations are becoming media organizations.

My reading of defensive mediatization here is old-school public relations recontextualized for a system where the news source also is a competitor. Fortunato (2001) and Suggs (2016), for instance, have shown how media access rules and policies are often macro-level choices that shape the interactions between the independent press and the front-line media staff or athletes. Indeed, the concept of steering probably means something different when the actors doing that work have a media production arm that competes with the independent press rather than just trying to influence the actions of the media. The point from McEnnis (302) suggests that the loss of monopoly control over mediated information is a major shift in the meaning of these traditional activities. Suddenly it is not just about managing relationships but is about competition. Defensive mediation plays into “jurisdictional disputes” (198). The phrase calls to mind boundary work perspectives in this area. Where I guess this leaves me is that these sports journalists simply do not see themselves as part of a system, but rather engaged in a personal drama with their interviewees and the teams. Perhaps this is a barrier to the commitment to serious journalism (565). Their inability to comment on the macro policies is a sign that the steering is especially effective.

I have a couple of smaller points to add. These are minor

L.47: In terms of the process of mediatization, what I think we are feeling here is that there is just so much more media and everything is new. Understanding what is supposed to what is a necessary process that takes significant time.

L:54: We would expect the news industry to be fully shaped by mediatization because they are media-centric institutions. I think what the authors mean is the mediatization of sports organizations.

L. 91: This is a good point and goes to this speaks to the nature of this media system. Journalistic exclusivity over storytelling was beneficial to the sporting institutions as well because of the limited control over information this arrangement provided. They might not know what journalists were going to write, or be able to prevent negative stories, but they could at least have only a few channels to monitor. The authors get at this point in the next paragraph.

L. 128: This probably is not the goal of the paper, but the authors should think about the ways this rearticulation of the media relations work done by sports organizations in the context of previous ways institutions deploy a media strategy. How much of this is new practice and how much of it is the adaptation of old practices to new sets of challenges?

I really enjoyed the paper and thought it presented rich data and could represent an important addition to the scholarly conversation in this area. 

Reviewer 2 Report

Thank you for the opportunity to review the paper, ‘Sports Organizations and their defensive mediatization strategies: The sports journalist’s perspective’. Overall, I found the paper to be interesting and thought-provoking. It discusses an important topic that has significant implications for the sports media industry, sports journalists, and sports fans.

The paper effectively employs Nölleke, Scheu, Birkner’s (2021) macro, meso, and micro levels to interpret and explain defensive mediatization strategies. However, I believe that the paper can be easily refocused to provide a more meaningful and impactful study that highlights the consequences of sporting organizations adopting defensive mediatization strategies. Specifically, the study should focus on the impacts and consequences of defensive media strategies on sports journalism and more broadly, how these strategies impact the type and quality of content generated and disseminated by the media, which also introduces consequences for fans and consumers.

In its current form, the study highlights how sports journalists are impacted by defensive mediatization strategies adopted by sport organizations, but what are the consequences of this? Why does it matter?

Currently the paper introduces the idea of mediatization and defensive mediatization strategies.  The overview of these concepts is adequate, but it could be reframed, with the focus reshaped to provide a more meaningful and impactful study that highlights the consequences of sporting organizations adopting defensive mediatization strategies.

Therefore, I suggest that the early sections of the paper should focus on, or at least introduce, theory and literature that highlights the implications of mediatization and defensive strategies of sporting organizations. This includes highlighting the consequences of sporting organizations creating their own media and bypassing independent media; athletes and officials providing dull, boring, beige interviews; sporting organizations acting in competition with independent media; social media metrics driving sensationalist narratives; athletes being interviewed by their own media teams; and public relations replacing news.

The introduction and literature review should introduce these questions and themes and refer to previous studies that have explored these issues. In doing so, the author/s will be able to create a narrative of why this study matters. What is lost, and what is gained when sporting organizations adopt defensive mediatization strategies?

The insights of the sports journalist should be positioned as enriching the theories and link to the previous work to highlight how the observations of the 16 journalists advance the existing body of work. The insights are interesting, but because the significance of their views is not as clear as what it could and should be, I suggest revisiting the discussion section to better tie the results to the theory and literature discussed in the early parts of the paper.

Additionally, the paper's current structure needs some improvement, as the results section seems to also include the discussion. Ideally, the discussion should discuss the results and what they mean, linking the results to the theory and literature discussed in the early parts of the paper.  Thus, I recommend refocusing the introduction and literature review as highlighted above, and then referring to the additions during the discussion.

Finally, I recommend replacing the outdated Hutchins and Rowe (2009) reference with a more contemporary source.

Round 2

Reviewer 2 Report

Congtratulations to the authors on their submission. The amendments undoubtedly make this a much stronger study.  The structure of the paper is much improved, meaning the significance of the research insights are now clearer, as is the research's contribution to the broader field.

I wish the authors well and look forward to seeing the published article soon.